# CDNF Interacts with ER Chaperones and Requires UPR Sensors to Promote Neuronal Survival

**DOI:** 10.3390/ijms23169489

**Published:** 2022-08-22

**Authors:** Ave Eesmaa, Li-Ying Yu, Helka Göös, Tatiana Danilova, Kristofer Nõges, Emmi Pakarinen, Markku Varjosalo, Maria Lindahl, Päivi Lindholm, Mart Saarma

**Affiliations:** Institute of Biotechnology, HiLIFE, University of Helsinki, 00790 Helsinki, Finland

**Keywords:** CDNF, neurotrophic factor, ER stress, unfolded protein response, dopamine neurons, sympathetic neurons, apoptosis, protein–protein interactions, mechanism of action

## Abstract

Cerebral dopamine neurotrophic factor (CDNF) is a neurotrophic factor that has beneficial effects on dopamine neurons in both in vitro and in vivo models of Parkinson’s disease (PD). CDNF was recently tested in phase I-II clinical trials for the treatment of PD, but the mechanisms underlying its neuroprotective properties are still poorly understood, although studies have suggested its role in the regulation of endoplasmic reticulum (ER) homeostasis and the unfolded protein response (UPR). The aim of this study was to investigate the mechanism of action of CDNF through analyzing the involvement of UPR signaling in its anti-apoptotic function. We used tunicamycin to induce ER stress in mice in vivo and used cultured primary neurons and found that CDNF expression is regulated by ER stress in vivo and that the involvement of UPR pathways is important for the neuroprotective function of CDNF. Moreover, we used AP-MS and BiFC to perform the first interactome screening for CDNF and report novel binding partners of CDNF. These findings allowed us to hypothesize that CDNF protects neurons from ER-stress-inducing agents by modulating UPR signaling towards cell survival outcomes.

## 1. Introduction

Cerebral dopamine neurotrophic factor (CDNF), a paralog of mesencephalic astrocyte-derived neurotrophic factor (MANF), was first characterized in 2007 as a trophic factor for midbrain dopamine neurons in a rat 6-hydroxydopamine (6-OHDA) model of Parkinson’s disease [1]. Together, CDNF and MANF form a CDNF/MANF protein family that is highly conserved in evolution and shows no sequence homology to other known protein families. Members of the CDNF/MANF protein family are small, approximately 18 kDa proteins that are highly soluble and monomeric in neutral solutions [1,2,3,4,5,6].

In animal models of different neuronal pathologies, CDNF and MANF promote the survival of various neuronal populations yet are structurally and functionally clearly distinct from classical neurotrophic factors (NTFs) [7,8,9]. NTFs such as proteins from the glial cell line-derived neurotrophic factor (GDNF) family of ligands (GFLs) and neurotrophins are secreted proteins that exert their survival-promoting effects by binding to the extracellular domains of their cognate plasma membrane receptors, thus activating intracellular signaling pathways, eventually leading to diverse cellular outcomes [10,11]. However, the impermeability of the blood–brain barrier (BBB) to proteins such as neurotrophic factors creates a clinical challenge, as NTFs have to be delivered to the brain directly to be able to bind their cognate receptors [8,12].

Although members of the CDNF/MANF protein family are partially secreted, data show that MANF resides in the lumen of the endoplasmic reticulum (ER) [13] and, to date, no transmembrane receptor for CDNF has been identified. Recent data demonstrate that a transmembrane protein, neuroplastin (NPTN), can serve as a cell surface receptor for MANF, modulating inflammatory responses and cell death [14]. Interestingly, CDNF was not able to bind to NPTN [14]. In addition to being an ER-resident protein, the expression and secretion of MANF are regulated by disturbances in the ER protein folding homeostasis, collectively known as the ER stress [13,15]. Such disturbances can be physiological—such as in pancreatic beta cells due to their production of high amounts of insulin [16,17]. Pathological ER stress occurs when cells are challenged by calcium depletion from the ER lumen, hypoxic conditions, or the accumulation of misfolded proteins in the ER lumen [18,19]. When cells undergo ER stress, they activate the unfolded protein response (UPR)—a collection of defense mechanisms aiming to restore ER homeostasis [20]. In mammalian cells, the UPR has three branches, each defined by an ER-resident transmembrane receptor: inositol requiring enzyme 1 (IRE1α), PKR-like ER kinase (PERK), and activating transcription factor 6 (ATF6) [20]. The regulated activation of these receptors aims to restore ER homeostasis by increasing the availability of protein folding factors and reducing the misfolded protein load. If cellular homeostasis is perturbed, UPR is initially cytoprotective. If this cytoprotective, adaptive UPR is unable to resolve ER stress, UPR shifts to an unresolved terminal UPR, leading to increased inflammatory signaling and cell death [21,22]. There is increasing evidence that the accumulation of aggregated/misfolded proteins, impairment of Ca^2+^ homeostasis, and dysregulation of axonal transport activate unresolved terminal UPR, possibly playing an essential role in the pathogenesis of neurodegenerative diseases such as Parkinson’s disease (PD) [23]. Therefore, approaches to control ER stress by preventing ER stress-induced apoptosis and promoting cell regeneration are important for the development of new therapeutic strategies for neurodegenerative diseases.

Several lines of evidence suggest that MANF can regulate UPR signaling [24,25,26]. It has been suggested that CDNF, too, is participating in the regulation of UPR signaling [27,28,29]. CDNF downregulated UPR markers in cultured DA neurons under ER stress and in the rat striatum in a 6-OHDA model of PD [27]. Based on in vitro studies using immortalized cells and hippocampal neurons, CDNF expression could induce an adaptive UPR and inhibit apoptotic pathways [27,29]. Moreover, the neuroprotective effects of CDNF in both in vivo and in vitro models of cerebral ischemia were suggested to occur through UPR pathways as the expression of CDNF was increased both by focal ischemia in the rat brain and in primary cultured neurons [30]. However, another study showed beneficial effects of CDNF against amyloid beta-induced synaptotoxicity and ER stress in primary cultured hippocampal cells [31]. However, direct molecular evidence showing the involvement of CDNF in UPR is virtually lacking.

The primary amino acid sequence of CDNF contains eight cysteine residues with conserved spacing—a characteristic feature of the CDNF/MANF protein family [1,2,3,4,32]. Structurally, CDNF and MANF are very similar, both consisting of two domains connected by a flexible linker region. The closest structural homologues to their amino-terminal (N-terminal), mainly alpha-helical, domains are proteins collectively known as the saposin-like proteins (SAPLIPs) [4,32]. The carboxy-terminal (C-terminal) domain of both CDNF and MANF contains two potentially important motifs. The first of these is a cysteine bridge formed from the CXXC motif. It has been shown that mutating the motif, ^127^CKGC^130^, abolishes the intracellular survival-promoting activity of MANF against ER stress in dorsal root ganglion neurons [33]. Although CDNF contains a highly similar motif, ^132^CRAC^135^ (Figure 1A), the importance of this for the survival-promoting activity of CDNF has not been addressed. Other potentially important motifs in the C-termini of human CDNF and MANF are the very C-terminal KTEL and RTDL sequences, respectively (Figure 1A). These motifs resemble the canonical KDEL sequence that functions as an ER retention signal for many ER-resident proteins. In line with this, several studies have shown that deletion of the RTDL motif causes MANF re-localization from the ER to the Golgi, as well as increasing the secretion of MANF [15,33,34]. In addition, removal of the KTEL sequence from CDNF was shown to increase CDNF secretion in vitro [35]. It has been suggested that the secretion of CDNF and MANF is regulated in a similar manner [35,36]. Overexpression of a KDEL receptor that interacts with KDEL-like sequences and functions in the retrieval of proteins from the Golgi to the ER decreased the secretion of both CDNF and MANF [36].

CDNF was recently shown to interact directly with alpha-synuclein (αSyn), a major protein of Lewy bodies characteristic for the neuropathology of PD, decreasing αSyn cell entry and reducing αSyn phosphorylation, as well improving locomotor activity in a rodent αSyn model of PD [37]. However, the exact mechanism and biological consequences of this interaction remain unclear. The role of CDNF in PD was further highlighted by recent studies as CDNF-deficient mice exhibited age-related loss of enteric neurons and functional alterations in the brain dopamine system—a phenotype reminiscent of early-stage PD [38,39]. Overall, the survival- and ER homeostasis-promoting mechanisms of CDNF are poorly understood. Here, we examined the role of UPR in the regulation and neuronal anti-apoptotic function of CDNF and provided in vivo evidence that CDNF is an ER stress-regulated protein. We found that CDNF does not act on naïve neurons but requires ER stress or injury to promote neuronal survival in vitro. We analyzed the effect that inhibiting UPR signaling has on the anti-apoptotic ability of CDNF and showed that inhibiting the IRE1α and PERK pathways of UPR blocks CDNF’s anti-apoptotic activity. Additionally, we performed the first protein–protein interaction screening for CDNF, and report novel protein partners of CDNF. We suggest that CDNF exerts its cytoprotective properties by modulating UPR signaling towards cell survival responses.

## 2. Results

### 2.1. Extracellularly Added CDNF Does Not Promote the Survival of Naïve Dopamine Neurons, but Rescues Thapsigargin-Treated Neurons from Apoptosis

CDNF was first characterized by its ability to protect and rescue midbrain dopamine (DA) neurons in vivo against 6-OHDA-induced neuronal damage [1]. To study the effects of extracellularly added CDNF on the survival of naïve DA neurons, and neurons under ER stress in vitro, we added CDNF to the culture media of mouse embryonic midbrain DA neurons. As a control, we used GDNF, which has been previously demonstrated to promote the survival of midbrain DA cultures [40]. CDNF, unlike GDNF, did not promote the survival of naïve DA neurons in culture (Figure 1B). A similar lack of survival-promoting effect on naïve DA neurons has previously been observed with MANF [25]. Next, ER stress and apoptosis were induced in DA neurons by adding thapsigargin (Tg), which selectively inhibits the sarco/endoplasmic reticulum Ca^2+^ ATPase (SERCA), causing disruption of ER Ca^2+^ homeostasis and activating the UPR [41]. Thapsigargin treatment reduced the survival of DA neurons by approximately half, whereas neurons treated with either CDNF or GDNF had an approximately 75% survival rate (Figure 1C). As we have previously demonstrated, CDNF, unlike GDNF, does not promote the survival of naïve neurons in vitro [1]. PD-inducing toxins such as 6-OHDA have been shown to specifically induce ER stress and activate the UPR in cultured neuronal cells in vitro [42]. In line with this, we show that CDNF can rescue 6-OHDA-lesioned DA neurons in culture (Figure 1D). Therefore, we conclude that the activity of CDNF relies on the stress-dependent activation or expression of its interaction partners, the identity of which remains undescribed.

### 2.2. Overexpressed CDNF Localizes to the ER and Protects Sympathetic Neurons against ER Stress-Induced Apoptosis

CDNF subcellular localization in neurons has not been characterized. Therefore, we next employed the superior cervical ganglion (SCG) neuron cultures to study the subcellular location of overexpressed CDNF in normal and ER stress conditions. For this, cultured SCG neurons were injected with expression plasmid encoding for full-length CDNF. CDNF immunofluorescence staining co-localized with PDI and GRP78 staining used to visualize the ER, but not with GM130 staining used as a marker for the Golgi complex (Figure 1E). Quantitative colocalization analysis using Pearson’s colocalization coefficient confirmed the notion that overexpressed CDNF is an ER-localized protein in SCG neurons, with the CDNF-PDI and CDNF-GRP78 Pearson’s coefficients being 0.44 and 0.55, respectively. The CDNF-GM130 Pearson’s coefficient was −0.03, indicating no colocalization between the CDNF and Golgi marker GM130 staining (Figure 1G). The intracellular localization of overexpressed CDNF remained unchanged upon treatment with tunicamycin (Figure 1F,H). We therefore concluded that while we could not detect endogenous expression levels of CDNF in mouse SCG neurons, overexpressed CDNF is localized to the ER. The ER-specific location of CDNF remained unchanged when ER stress was induced with tunicamycin. These data suggest that CDNF is an ER-localized protein in neurons.

While extracellularly applied CDNF has been shown to be both neuroprotective and neurorestorative in animal models of neuronal damage [1,30,37,43,44], the intracellular effects of CDNF remain relatively poorly studied. Therefore, we used CDNF overexpression by neuronal plasmid microinjection to investigate its effects on neuronal survival in an ER stress-related apoptosis model in vitro. For this, we treated sympathetic SCG neurons with tunicamycin (Tm), which is an inhibitor of N-linked glycosylation in the ER, causing the accumulation of improperly processed glycoproteins in the ER, resulting in the activation of the UPR and, finally, apoptosis [2,25,45,46,47]. As expected, treating the SCG sympathetic neurons with tunicamycin reduced their survival by around 70% (uninjected Tm+ vs. uninjected Tm, Figure 1I,J). Overexpressed CDNF had a survival-promoting effect, as demonstrated by the increased survival of SCG neurons injected with CDNF-expressing plasmid compared to neurons injected with vector only (Figure 1I,J). We also assessed the importance of two conserved sequence motifs to the antiapoptotic activity of CDNF. For this, we mutated the C-terminal CXXC motif ^132^CRAC^135^ to ^132^CRAS^135^ (CDNF C135S) or deleted the very C-terminal KTEL-sequence (CDNF ΔKTEL) and injected SCG neurons with the respective plasmids (Figure 1A,I,J). Mutating the CXXC motif in the C-terminal domain of CDNF reduces but does not completely abolish the antiapoptotic effect of CDNF in ER-stressed SCG neurons (Figure 1I). Deleting the very C-terminal ER-retention signal-like KTEL-sequence abrogated the antiapoptotic effect of CDNF (Figure 1J).

### 2.3. CDNF Is an ER Stress-Induced Protein In Vivo

Upregulation of CDNF expression upon ER stress induction by tunicamycin treatment in hippocampal neurons has been demonstrated in vitro [30]. However, whether the expression of CDNF is induced in ER stress conditions in vivo has not been investigated. We therefore aimed to study whether the expression of CDNF is changing in an in vivo model of ER stress. For this, we injected mice intraperitoneally with tunicamycin and analyzed the expression levels of CDNF and typical UPR genes in the brain, kidney, and liver taken 48 h after the injection. Increased expression of several UPR pathway transcripts confirmed ER stress in all tissues analyzed (Figure 2A–C). However, tissues were responding differently to tunicamycin, as indicated by the differential activation of UPR pathways. All three pathways, IRE1α, PERK, and ATF6, were activated in the liver and kidney (Figure 2B,C), whereas only IRE1α was activated in the brain, as indicated by the mild upregulation of *sXbp1* and *Grp78* (Figure 2A). Additionally, upregulation of the *Chop* transcript in the liver and kidney indicates the activation of the proapoptotic branch of PERK downstream signaling in the peripheral tissues (Figure 2B,C).

Analysis of CDNF mRNA and protein levels in the tissues of Tm-injected mice showed that CDNF is an ER stress-regulated protein. More specifically, we observed *Cdnf* mRNA being upregulated in both kidney and liver tissues and CDNF protein upregulated in the liver 48 h after administering Tm to mice (Figure 2F,G). Therefore, we show, for the first time, that Tm administration in vivo in mice leads to an increase in CDNF mRNA and protein levels, disclosing CDNF as an in vivo ER stress-inducible protein. MANF, being an ER stress-regulated protein, has been found upregulated in vivo in different mouse tissues after Tm administration [48,49]. In accordance, *Manf* mRNA was upregulated in liver and MANF protein in kidney and liver tissues 48 h after administering Tm to mice (Figure 2D,E). We did not observe any statistically significant upregulation of CDNF and MANF mRNA or protein in the brain, reflecting the mild ER stress in the adult mouse brain after intraperitoneal Tm injection, most likely arising from the lower concentration of the drug in the brain as compared to the periphery (Figure 2D–G).

### 2.4. Extracellularly Applied CDNF Regulates UPR Signaling Pathways in ER-Stressed Dopamine Neuron Cultures

CDNF has been increasingly associated with the ER stress response and UPR regulation [27,29]. Recombinant CDNF protein has been shown to reduce levels of the *Atf6* UPR transcript in thapsigargin-treated embryonic mouse dopamine neurons in vitro and the level of GRP78 protein and ATF6 transcripts in the rat striatum in a 6-OHDA model of Parkinson’s disease in vivo [27]. In the current study, we have shown that extracellularly applied CDNF rescues DA neurons from ER stress-induced apoptosis in vitro. To gain a better understanding of how CDNF rescues cultured DA neurons from UPR-induced apoptosis, we analyzed the expression of a selection of UPR-related genes corresponding to all three major UPR pathways. Transcriptionally active spliced *Xbp1* (*sXbp1*) directly corresponds to the activation of the IRE1α pathway, participating in the regulation of other genes downstream from the IRE1α receptor, such as *Grp78*. In addition, thapsigargin-induced ER stress upregulates the expression of *Atf4* and *Chop* in the PERK pathway and *Atf6* in the ATF6 pathway.

As expected, treatment with thapsigargin significantly increased the mRNA levels of all UPR genes studied in cultured DA neurons (Tg vs. control, Figure 3A–F). When CDNF was applied together with Tg, it was able to significantly reduce the expression of *sXbp1, Atf6*, and *Grp78* transcripts (Tg vs. Tg + CDNF, Figure 3B,E,F). While the *Atf4* and *Chop* transcripts were also reduced in CDNF-treated cultures, they did not reach statistical significance. We, therefore, conclude that extracellularly applied CDNF rescues DA neurons from ER stress-induced apoptosis through regulating the IRE1α and ATF6 branches of UPR signaling after 24 h.

### 2.5. The Anti-Apoptotic Effect of Intracellularly Applied CDNF Relies on the Activity of IRE1α and PERK Pathways

In order to further investigate the involvement of UPR signaling in the antiapoptotic effect of CDNF in neurons, we used our SCG neuron in vitro model of ER stress-related neuronal apoptosis. For this, we delivered CDNF intracellularly by microinjecting CDNF protein into SCG neurons before inducing ER stress with tunicamycin. Different UPR signaling pathways in neurons were dampened by adding either PERK inhibitor GSK2606414 [50] or IRE1α RNase inhibitor 4μ8C [51] or IRE1α kinase inhibitor KIRA6 [52] to the culture medium. CDNF protein protected SCG neurons as compared to Tm-treated controls (Figure 3G). This protective effect was diminished when either PERK or IRE1α signaling was inhibited by adding any of the inhibitors. We therefore concluded that the anti-apoptotic effect of overexpressed CDNF is dependent on the activity of both IRE1α and PERK pathways.

### 2.6. Characterization of CDNF Expression Constructs for Affinity Purification Coupled to Mass Spectrometry

While understanding the protein–protein interactions (PPIs) is key to understanding the molecular function of a protein, very little is known about the molecular interactions of CDNF. Therefore, we set out to characterize the PPIs of CDNF using affinity purification coupled to mass spectrometry (AP-MS). The workflow of AP-MS is presented in Figure 4A. To allow for subsequent streptavidin affinity purification, an SH tag consisting of Twin-StrepII and HA tags was inserted between the pre- and mature regions of the human CDNF expression construct (Figure 4A). The pre-SH-CDNF construct was as efficiently protecting SCG neurons from tunicamycin-induced apoptosis as the untagged wild-type CDNF expression construct, indicating that the SH-tag does not interfere with the biological activity of CDNF (pTO pre-SH-CDNF vs. pCR3.1 CDNF; Figure 4B). Next, the pre-SH-CDNF expression construct was used to generate doxycycline-inducible cell lines originating from either Flp-In T-REx^TM^ HEK293 (HEK293 parental, Invitrogen, Waltham, MA, USA) or Flp-In T-REx INS1 #5-3.19 (INS1 parental, [53]). Both parental cell lines contain a single genomic Flp-recombinase target (FRT) site, allowing for easy Flp-mediated generation of stable isogenic cell lines. The cell lines originating from INS1 parental or HEK293 parental cell lines and inducibly expressing SH-tagged CDNF were named INS1 SH-CDNF and HEK293 SH-CDNF, respectively. Additionally, an SH-tagged GFP construct was used to create INS1 SH-GFP and HEK293 SH-GFP cell lines to be used as controls in subsequent AP-MS analysis. Finally, we checked for the doxycycline-inducible expression of SH-tagged CDNF (SH-CDNF) from INS1 and HEK293 parental and INS1 SH-CDNF and HEK293 SH-CDNF cell lines (Figure 4C). We did not detect endogenous levels of CDNF from any of the cell lysates analyzed using Western blotting, but in cells treated with doxycycline, we did detect CDNF antibody reactive bands, with their size approximately corresponding to CDNF with SH tag (Figure 4C). Therefore, having confirmed that the SH-tagged CDNF is biologically active and inducibly expressed from the INS1 SH-CDNF and HEK293 SH-CDNF cell lines, we continued with the AP-MS.

### 2.7. CDNF Interactomes in HEK293 and INS1 Cell Lines

We used the SH-tagged CDNF and SH-tagged GFP from the respective HEK293 and INS1 cell lines as bait for the streptavidin affinity purification of protein complexes and subsequent MS analysis. To efficiently filter out the contaminating background proteins, the base2 logarithmized fold change ratios of each protein in SH-CDNF/SH-GFP cell lines were plotted against their respective log10 (*p*-values). Significantly enriched hits from both cell lines are shown on the right side of the respective volcano plot parabolas (Figure 5A,B). As a result, we observed 65 high-confidence PPIs in the CDNF interactome in HEK293 cells (Figure 5A,C) and 30 PPIs in the CDNF interactome in INS1 cells (Figure 5B,D), excluding CDNF itself.

In order to obtain a better understanding of the biological role of CDNF, we performed Gene Ontology Biological Process (GO BP) term overrepresentation analysis of the CDNF interactomes from both HEK293 and INS1 cell lines using the Panther online platform (http://pantherdb.org/, accessed on 17 November 2020). Statistically significantly (*p* < 0.05) overrepresented GO BP terms were arranged in the decreasing order of fold enrichment. The five most enriched GO BP terms from the HEK293 SH-CDNF interactome were biotin metabolic process, viral RNA genome packaging, regulation of mitotic spindle assembly, chaperone cofactor-dependent protein refolding, and protein refolding (Figure 5E). The five most enriched GO BP terms from the INS1 SH-CDNF interactome were early endosome to Golgi transport, positive regulation of DNA binding, response to unfolded protein, maintenance of location, and response to endoplasmic reticulum stress (Figure 5F). The cell lines studied for the CDNF interactome differ in both the organism and the tissue that they originate from—the INS1 cell line is derived from rat insulinoma and used as a model for pancreatic islet beta cell function, and HEK293 is derived from the human embryonic kidney. It is, therefore, interesting to note that the only GO BP terms significantly overrepresented in both the INS1 and HEK293 CDNF interactomes are related to the cellular response to unfolded protein and the response to endoplasmic reticulum stress. This suggests that the most conserved biological role of CDNF is related to those processes involved in regulating ER protein homeostasis. This agrees with our observation about the involvement of UPR signaling in the survival-promoting action of CDNF against ER stress in DA and SCG neurons in vitro.

### 2.8. The ER-Localized Interactome of CDNF

Since CDNF is intracellularly localized mostly to the ER lumen (Figure 1D–G, [35]), we decided to investigate the ER-localized interactome of CDNF more closely. For this, we assembled an interaction network consisting of a subset of CDNF interactomes combined from both INS1 SH-CDNF and HEK293 SH-CDNF cell lines and consisting of proteins with ER localization (ER lumen or ER membrane), ER-associated localization (secreted or mitochondrial outer membrane protein), or ER-associated posttranslational modifications, such as N-linked glycosylation, annotated in the UniProtKB database (https://www.uniprot.org/, accessed on 4 December 2020). In total, the ER-localized interactome of CDNF consisted of 21 proteins (Figure 6A). Of these, seven proteins had ER luminal localization similar to CDNF. These proteins were: GRP78, glucose-regulated protein 170 kDa (GRP170, also known as HYOU1 and ORP150), protein disulfide-isomerase (PDIA1), protein disulfide-isomerase 6 (PDIA6), neutral alpha-glucosidase AB (GANAB), FAD-dependent oxidoreductase domain-containing protein 2 (FXRD2), and prolyl 3-hydroxylase 1 (P3H1). Only two proteins, GRP78 and GRP170, were detected as CDNF interactors in both INS1 and HEK293 cell lines. GRP78 and GRP170 have previously been reported as interacting with MANF [15,25,54], but their interaction with CDNF has not been reported. Therefore, we used the split Venus-based bimolecular fluorescence complementation assay (BiFC) to further verify the interaction of GRP78 and GRP170 with CDNF [25]. The transcription factors Jun and Fos [55] and ER proteins GRP78 and GRP170 [25] were used as positive controls for BiFC signal formation in the nucleus and ER, respectively (Figure 6B). Non-interacting transcription factors Jun and Max served as a negative control of BiFC signal formation. In our analysis, CDNF gave a positive BiFC signal with both GRP78 and GRP170 (Figure 6B). The BiFC signal of CDNF and GRP170 and CDNF and GRP78 colocalized with the anti-calreticulin immunostaining used to visualize the ER, indicating that both interactions of CDNF take place in the ER (Figure 6B). All in all, we here show that CDNF interacts with both GRP78 and GRP170 in the ER. Since GRP170 is a co-factor of GRP78, preferentially interacting with the ADP-bound form of GRP78 [54], it is possible that CDNF interacts with the GRP78-GRP170 complex, rather than individual proteins alone.

### 2.9. C-Terminal Domain of CDNF (C-CDNF) Preferentially Interacts with the Nucleotide-Binding Domain of GRP78

CDNF and GRP78 are both two-domain proteins. The distinct functions of the N-terminal nucleotide-binding domain (NBD) and the C-terminal substrate-binding domain (SBD) of GRP78 have been well described [56]. Similarly, the two domains of CDNF have been proposed to have distinct roles in the molecular biology of CDNF, although experimental data about these possible roles are still lacking [7,9,57].

Since the interaction or the purpose of this interaction between GRP78 and CDNF has not been reported before, we next investigated which domain of GRP78 CDNF interacts with. For this, we cloned the NBD (amino acids 1–405 and 648–654) and SBD (amino acids 1–18 and 414–654) of GRP78 to a BiFC vector. Both constructs contained the ER signal sequence as well as the C-terminal ER retention sequence to facilitate the correct intracellular localization of NBD and SBD of GRP78 to the ER. Similarly, we constructed BiFC vectors expressing the N-terminal domain of CDNF (N-CDNF; amino acids 1–100) and C-CDNF (amino acids 101–161). As a positive control for BiFC signal formation, we used full-length CDNF and GRP78. Analysis of BiFC signal formation revealed that full-length GRP78 formed contacts with CDNF, N-CDNF, and C-CDNF. Similarly, full-length CDNF gave a positive BiFC signal with full-length GRP78 and its SBD and NBD, although the signal was more diffuse and weaker with the GRP78 domains, especially with SBD, than with the full-length GRP78 construct. Importantly, while N-CDNF and C-CDNF were able to give a strong positive BiFC signal with full-length GRP78, neither gave a positive signal with the SBD. A particularly strong BiFC signal between individual domains was observed with C-CDNF and NBD of GRP78 (Figure 6C). All in all, these data indicate that while both domains of CDNF and GRP78 may participate in the interaction between CDNF and GRP78, the main interaction interfaces lie within the NBD of GRP78 and C-terminal domain of CDNF. Moreover, while additional experiments are clearly needed, these data also suggest that CDNF is a cofactor rather than a substrate of GRP78.

## 3. Discussion

Although CDNF has been tested in phase I-II clinical trials in Parkinson’s disease patients [8,9], the molecular mechanism of its action is still under investigation. Despite recent publications associating CDNF with the UPR [27,29], the roles for CDNF in ER stress in vitro and in vivo have not been widely explored. In the current study, we demonstrate for the first time that CDNF is an ER stress-responsive protein in vivo. In vitro, using primary cultures of DA and SCG neurons, we show that both extracellularly and intracellularly delivered CDNF promotes neuronal survival against ER stress. To understand the potential molecular mechanisms of CDNF’s action, we characterized CDNF interactions with cellular proteins using high-throughput protein–protein interaction screening in HEK293 and INS1 cell lines and identified novel ER luminal partners for CDNF, including GRP78 and GRP170. Our results are in support of the function of CDNF in the regulation of UPR in cells and neurons to promote their survival.

We show that, unlike neurotrophic factor GDNF protecting DA neurons from apoptosis [58], CDNF does not promote the survival of naïve DA neurons in culture. Instead, CDNF requires ER stress conditions to be able to exert its antiapoptotic properties on DA neurons in vitro. The same has been suggested to be true for MANF as well [25,59], despite its initial discovery as a neurotrophic factor for naïve DA neurons [6,25,59]. We therefore hypothesize that the cellular counterparts responsible for carrying out the survival-promoting function of extracellularly added CDNF and MANF become either activated or upregulated upon conditions of cellular stress. The nature of these counterparts remains unclear as of yet, as no binding to plasma membrane receptors or uptake by endocytosis has been reported for CDNF. Since saposin-like proteins usually interact with lipids or membranes, it was proposed that the N-terminal domain mediates the CDNF/MANF interaction with lipids [57]. Recently, MANF was shown to bind sulfoglycolipid 3-*O*-sulfogalactosylceramide (sulfatide), and the presence of sulfatide was shown to facilitate the cellular uptake and cytoprotective effect of extracellularly added MANF [60]. While CDNF does not bind sulfatide [60], the presence of the SAPLIP domain suggests that CDNF, too, may still use interactions with other types of lipids to enter cells. Recent biochemical data show that neuroplastin (NPTN) can serve as the cell surface receptor for MANF and, through this receptor, modulate inflammatory responses and counteract cell death [14]. Interestingly, CDNF was not able to bind to NPTN [14]. We used two different ER stress-inducing agents—thapsigargin and tunicamycin—to show the ability of CDNF to rescue two neuronal cell types from apoptosis.

Using in vitro cultures of DA and SCG neurons, we show that CDNF is an ER-localized protein that can regulate UPR signaling and rescue neurons from ER stress-induced apoptosis. In SCG neurons treated with tunicamycin, CDNF expression plasmids were delivered intracellularly by microinjection. In this paradigm, wt CDNF and, to some extent, its C135S mutant, but not its ΔKTEL mutant, were able to promote the survival of tunicamycin-treated SCG neurons. Both CDNF and MANF have a KDEL-like ER retention sequence in their very C-termini. Deleting this ER retention signal RTDL of MANF rendered MANF partially mislocalized from the ER to the Golgi complex and, consequently, unable to promote the survival of etoposide-treated SCG neurons [33]. Additionally, in the fruit fly, deletion of the putative ER retention signal RSEL of DmManf decreased its ER localization and consequently its ability to rescue the larval lethality of DmManf mutants [61]. Deleting the KTEL sequence improved CDNF secretion from stable transgenic CDNF-expressing ARPE-19 as compared to the wt CDNF-expressing ARPE-19 cell clone [35]. Moreover, deleting the very C-terminal QTEL amino acids of the rat CDNF expression construct was shown to increase the secretion and reduce the intracellular levels of the corresponding overexpressed CDNF ΔQTEL protein from HEK293-T cells. As a result, the CDNF ΔQTEL construct failed to promote the survival of HEK293-T cells treated with thapsigargin [29]. Therefore, it is likely that the loss of anti-apoptotic activity observed for CDNF ΔKTEL in our study is a consequence of its reduced ER localization. Importantly, a recent study identified CDNF as a novel cardiomyokine that promotes cardioprotection via binding to the KDEL receptor and PI3K/AKT activation [62]. We therefore conclude that CDNF, similar to MANF, needs to be localized to the ER to exert its cell survival-promoting role.

CDNF is characterized by eight cysteine residues with conserved spacing. These cysteine residues form four intramolecular disulfide bridges: three in the N-terminal domain and one in the C-terminal domain. The C-terminal disulfide bridge of both CDNF and MANF forms a CXXC motif that is also commonly found in the catalytic centers of redox enzymes [63]. Despite this, MANF was found not to have oxidoreductase activity [33,64]. Mutating the CXXC motif of CDNF (CDNF C135S) reduced but did not completely abolish the ability of CDNF to counteract tunicamycin-induced neuronal apoptosis. This is unlike similar studies with MANF, where the CXXC motif of MANF was shown to be essential for DmManf functionality [61] and important for the neuroprotective activity of MANF in vitro against ER stress and in an in vivo model of stroke [33]. We hypothesize that the importance of the CXXC motif for the neuroprotective activity of CDNF arises from its possible role in stabilizing CDNF into a conformation favoring anti-apoptotic protein–protein interactions. We conclude that the CXXC and KTEL ER retention signal motifs are important for the anti-apoptotic activity of CDNF, but more studies are needed to elucidate the exact functions of these motifs.

Previously published in vitro studies have established a hypothesis that CDNF regulation in ER stress may be cell type-dependent. For example, an ER stress-induced increase in CDNF expression was detected in cardiomyocytes [65], but not in an osteosarcoma-derived cell line [26]. Moreover, a recent study showed that while CDNF and MANF show redundancy in some mouse tissues, they still have some different target tissues where they exert their function as ER stress regulators [28]. The results of our study, for the first time, revealed that endogenous CDNF levels were increased in response to ER stress-inducing intraperitoneal injections of tunicamycin in the mouse liver and kidney in vivo. Similarly, increased levels of MANF in mouse tissues have been reported after ER stress induction in vivo [48,49], Therefore, our results indicate that CDNF expression is regulated by ER stress in mouse tissues.

Using ER-stressed cultured DA neurons, we studied the anti-apoptotic effects of extracellularly added CDNF. QPCR analysis of UPR genes indicated that the survival-promoting activity of CDNF in DA neurons is related to its ability to downregulate UPR signaling. Thapsigargin treatment in DA neurons increased the transcript levels of all UPR genes studied, and although the addition of CDNF to the culture medium was able to downregulate all transcripts tested, only *sXbp1*, *Atf6*, and *Grp78* transcripts were significantly downregulated. This suggests that, in this model of ER stress-related apoptosis of DA neurons, CDNF is able to regulate at least the IRE1α and ATF6 UPR signaling pathways.

We also used intracellularly delivered CDNF to investigate the role of UPR signaling in the anti-apoptotic activity of CDNF in cultured SCG neurons. When PERK or IRE1α signaling was inhibited using GSK2606414 [50], KIRA6 [52], or 4μ8C [51] reagents, CDNF was no longer able to rescue SCG neurons from tunicamycin-induced apoptosis. This was observed regardless of whether CDNF was overexpressed from a plasmid microinjected into the cell nucleus or recombinant CDNF protein was directly microinjected into the cytoplasmic space of SCG neurons. Two conclusions can be made from this observation. First, the survival-promoting mechanism of CDNF requires the activity of both PERK and IRE1α signaling pathways, at least in this in vitro model of ER-stress related apoptosis of SCG neurons. Second, since the survival-promoting effect in SCG neurons was similar regardless of whether CDNF was overexpressed from a plasmid or delivered to the cytoplasm in excess amounts as a recombinant protein, the survival-promoting action of CDNF in this model is likely intracellular and does not occur in a paracrine or autocrine fashion. The details of the molecular machinery leading to this PERK- and IRE1α-mediated intracellular survival-promoting activity of CDNF remain to be studied.

It is unclear why, in SCG neurons, the activity of PERK and IRE1α pathways was needed for the survival-promoting activity of CDNF, but, in DA neurons, only transcripts representing the activation of IRE1α and ATF6 pathways were significantly downregulated upon CDNF application. Another recent study reported the protective effect of transiently overexpressed CDNF against thapsigargin-induced apoptosis in HEK293-T cells [29]. Interestingly, differently from our study, the survival-promoting effect of CDNF arose from its ability to upregulate transcripts and proteins representing the activation of all three UPR branches [29]. One possible explanation for this discrepancy between the previous study and ours is the fact that UPR is an adaptive response, the extent, temporal sequence of events, and cellular outcome of which depends, among other things, on the underlying cause of the stress, the biological model used, and the phase at which the response is being studied [66,67,68]. DA and SCG neurons are structurally and functionally different. Additionally, we have used extracellular CDNF delivery to study its effects on thapsigargin-induced ER calcium disbalance in DA neurons and intracellular CDNF delivery to study its effects on ER stress caused by the accumulation of misfolded glycoproteins in SCG neurons. Therefore, it seems that as different ER stressors applied on different cell types initially launch the activation of different cellular pathways that later converge to ER stress-related apoptosis, CDNF can flexibly counteract the detrimental effects of different stressors on several cell types. We therefore hypothesize that, in our SCG neurons treated with tunicamycin, the protective effect of CDNF arises from its ability to induce an early adaptive UPR response through the PERK and IRE1α pathways. Correspondingly, our results with cultured DA neurons suggest that, in this model, CDNF was able to rescue neurons from thapsigargin-induced apoptosis through dampening the chronic later-stage pro-apoptotic UPR signaling. Our data, therefore, suggest that CDNF is a master regulator of UPR that can modulate several branches of UPR signaling throughout different stages of cell fate.

Information about the interactome of a protein provides important clues for understanding its molecular mechanism and function. Thus far, knowledge about the protein–protein interactions of CDNF has been lacking. PPI research relies on several very useful methods for studying PPIs, such as protein complementation assays, protein microarrays, and bioluminescence or Förster resonance energy transfer (BRET or FRET) [69,70,71,72]. One widely preferred method for interaction proteomics is the affinity purification coupled to mass spectrometry that we have previously successfully used [25,73]. We therefore went on to screen for the protein–protein interactions of CDNF using the AP-MS. For our interactome screening, we used two mammalian cell lines originating from different species and tissues: the human embryonic kidney-derived HEK293 and rat pancreatic insulinoma-derived INS1 cell line. While neither of these cell lines is neuronal, both have been well established for the generation of inducible expression cell lines for PPI screening purposes [25,53,73,74]. Additionally, CDNF mRNA and protein have been detected from a wide range of tissues studied, including kidney and pancreatic tissue [75]. Given the conserved sequence and widespread expression of CDNF, the HEK293 and INS1 cell lines provide a good starting point for the screening of PPI of CDNF. We report that the CDNF interactome in HEK293 and INS1 cells consists of 65 and 30 proteins, respectively. GO term overrepresentation analysis of the CDNF interactomes revealed several biological processes related to the proteins in the CDNF interactome. However, the ER stress and unfolded protein GO terms represented the only GO terms significantly overrepresented in both cell lines studied. This suggests that the CDNF-interacting proteins behind these GO terms represent the most conserved interactions of CDNF and, therefore, the most conserved biological function of CDNF is related to these processes. The role of CDNF in the protein folding and homeostasis processes in the ER has started to emerge [27,29,30], supporting our in vivo and in vitro data, and validating our AP-MS results.

Since the AP-MS captures protein complexes, it is unclear how many of the proteins of the CDNF interactome represent proteins interacting with CDNF directly. Based on our current study and previously published data, CDNF is intracellularly localized to the ER lumen and is, thus, more likely to be interacting with other proteins having ER localization [29,35,36]. We therefore took a closer look at the ER-localized interaction subproteome of CDNF. For this, the subcellular locations of CDNF interacting proteins from both HEK293 and INS1 cells were determined based on their respective UniProtKB database entries, as well as the available literature about each protein. This revealed a total of 21 proteins forming what we termed the ER-localized interactome of CDNF. Out of these, seven proteins are bona fide ER luminal proteins and thus most likely represent directly interacting proteins. We decided to further verify the interactions of ER luminal proteins that were identified as CDNF interactors in both HEK293 and INS1 cell lines studied. These proteins were ER chaperones GRP78 and GRP170. Neither has been reported as an interaction partner for CDNF before, but both have been shown to interact with MANF [15,25,54]. Recently, MANF was shown to be a nucleotide exchange inhibitor of GRP78 that, by preventing ADP release from and ATP binding to GRP78, aids substrate folding by GRP78 [54]. However, the interaction with GRP78 is not essential for the survival-promoting action of MANF [25]. Both GRP78 and GRP170 gave a positive BiFC signal with CDNF, thus verifying the interaction also in the cells. Finally, we demonstrate that, in a cellular context, CDNF preferentially binds the N-terminal nucleotide-binding domain of GRP78 through its C-terminal domain. These data suggest that CDNF, as with MANF and GRP170, is a cofactor of GRP78. Further studies should establish whether CDNF binds UPR sensors and ER folding machinery as MANF does [25,76].

To conclude, we have demonstrated that CDNF is a bona fide UPR protein, being upregulated by and possessing the ability to modulate UPR signaling towards favorable cellular outcomes. Furthermore, functioning IRE1α and PERK pathways are required for the antiapoptotic action of CDNF in neurons.

## 4. Materials and Methods

### 4.1. Mice and Treatment with Tunicamycin

Mice for Tm treatment were maintained on an ICR (CD-1, Envigo, RRID:MGI:5658486) background. All mice, including NMRI mice and TH EGFP transgenic mice (Envigo, RRID:MGI:2160177) [77,78], were housed under a 12-h dark/light cycle with *ad libitum* access to water and food. This animal study was conducted in accordance with the Declaration of Helsinki and approved by the Regional State Administrative Agency for Southern Finland, protocols ESAVI/10564/04.10.07/2014, approved 15.1.2015, and ESAVI/8189/04.10.07/2015, approved 11.11.2015. Seven-week-old mice were intraperitoneally injected with 150 μL of PBS containing 1 mg/kg/mouse of tunicamycin (Abcam, Cambridge, MA, USA) first dissolved as 2 mg/mL in dimethyl sulfoxide (DMSO). Control mice were injected with 150 μL of a 10% (*v*/*v*) dilution of DMSO in PBS corresponding to an equivalent volume of DMSO injected to TM-treated mice. Mice were sacrificed 48 h after the injection and tissues were collected for analysis.

### 4.2. RNA Isolation, Reverse Transcription Quantitative PCR (RT-qPCR)

Mice were euthanized using carbon dioxide and tissues were collected and fast-frozen in liquid nitrogen, followed by RNA isolation, reverse transcription, and RT-qPCR as described previously [24]. The expression levels were normalized to the β-actin levels in the same samples. Primer sequences for *Grp78*, *Chop*, *sXbp1*, *tXbp1*, *Atf4*, *Atf6α*, *Atf6β*, *Manf*, *Cdnf* transcripts used in this study have been described before [24,25,75,79].

### 4.3. Protein Extraction and Enzyme Linked Immunosorbent Assay (ELISA)

Mouse tissue samples were homogenized as described previously [75]. The levels of endogenous CDNF and MANF in mouse tissue lysates were analyzed by in-lab-developed and validated sandwich m(mouse) CDNF and mMANF ELISAs [75,80]. The dynamic range of mCDNF ELISA was 15.6–500 pg/mL and sensitivity 6 pg/mL. The mCDNF ELISA specifically recognized mouse CDNF in tissue lysates as values from *Cdnf*^−/−^ mouse samples remained below the sensitivity level [75]. In brief, for mCDNF ELISA, MaxiSorp (Nunc) microtiter plates were coated with 1 μg/mL goat anti-mCDNF antibodies (AF5187, R&D Systems, Minneapolis, MN, USA) in 0.05 M carbonate/bicarbonate buffer (pH 9.6). Samples and the standard (recombinant mCDNF, 5187-CD, R&D Systems) were diluted in blocking buffer (3% BSA in PBS) and incubated on the plate overnight at 4 °C. On the next day, rabbit anti-CDNF antibodies [1] at 0.2 μg/mL in blocking buffer were incubated on the plate for 3 h at 37 °C, followed by incubation with horseradish peroxidase (HRP)-conjugated donkey anti-rabbit IgG secondary antibodies (GE Healthcare, Chicago, IL, USA). Antibody complexes were visualized using 3,3′,5,5′-tetramethylbenzidine (DuoSet ELISA Development System, R&D Systems) and absorbance was read at 450 and 540 nm (Victor3 reader, Perkin Elmer, Waltham, MA, USA).

The dynamic range of mMANF ELISA was 62.5–1000 pg/mL and its sensitivity was 29 pg/mL [80]. mMANF ELISA recognized both mouse and human MANF. It did not give a signal from *Manf*^−/−^ tissue lysates, indicating that it specifically recognized MANF in mouse tissue samples [80]. mMANF ELISA was built using goat antibodies to MANF (AF3748, R&D Systems) for coating, rabbit anti-MANF antibodies (LS-B2688, LSBio, Seattle, WA, USA) for detection, and HRP-linked secondary antibodies against rabbit IgG [80]. Mouse tissue lysates and the standard (recombinant human MANF, P-101–100, Icosagen) were diluted in blocking buffer of 1% casein in PBS–0.05% Tween 20 before application to the plate [80].

Protein concentrations in the tissue lysates were determined using the DC™ Protein Assay Kit I (500-0111, Bio-Rad, Hercules, CA, USA). For mCDNF ELISA, mouse brain, kidney, and liver lysates were diluted at 1:40, 1:100, and 1:10, respectively, in blocking buffer. For mMANF ELISA, brain tissue lysate was diluted at 1:4000, and kidney and liver lysates at 1:5000 and 1:10,000, respectively, in blocking buffer. Levels of CDNF and MANF were normalized to the total protein concentration of a sample. All measurements were done in duplicate.

### 4.4. CDNF Expression Plasmids

The generation of pCR3.1 CDNF and pCR3.1 CDNF delKTEL has been described elsewhere [1,35]. pCR3.1 CDNF C134S was generated similarly to pCR3.1 CDNF delKTEL—by site-directed inverse PCR mutagenesis with pCR3.1 CDNF as a template, and with primers 5′-GCAGGGCCTCTGCAGAAAAAAC-3′ and 5′-ACTCCTCCCCCCAGCTATG-3′.

### 4.5. Generation of Stable Isogenic Doxycycline-Inducible Cell Lines and CDNF Immunoblotting

To generate the CDNF Gateway compatible entry vector, CDNF from pCR3.1 was cloned into the pENTR221 vector using the Gateway protocol for entry clone generation by PCR (Invitrogen, Thermo Fisher Scientific, Waltham, MA, USA). The Twin-Strep- and a hemagglutinin- (SH) tag were amplified from the pcDNA5/FRT/TO/SH vector and inserted between the sequences coding for the signal peptide (pre-) and mature regions of human CDNF to generate the pre-SH-CDNF entry clone. Finally, the pre-SH-CDNF was cloned into the pcDNA5/FRT/TO/cSH destination vector using Gateway LR cloning (11791020, Invitrogen, Thermo Fisher Scientific). This construct was then used for activity testing and generating stable isogenic doxycycline-inducible pre-SH-CDNF cell lines. The SignalP 4.0 signal peptide prediction algorithm [81] was used to check that the CDNF ER signal peptide would still be recognized as such and thus cleaved by the mammalian signal peptidase complex. Flp-In 293 T-REx™ cells (HEK293 parental, R78007, Invitrogen) containing a single genomic FRT site and stably expressing the Tet repressor were grown in Dulbecco’s modified Eagle medium (DMEM, Sigma-Aldrich, St. Louis, MI, USA) supplemented with 10% fetal bovine serum (Gibco, Thermo Fischer Scientific) and 50 µg/mL Normocin (ant-nr-2, Invivogen, San Diego, CA, USA). Flp-In INS1 #5-3.19 cells (INS1 parental) were a gift from G. Ryffel and S. Senkel and have been described elsewhere [53]. INS1 parental cells were grown in RPMI-1640 media supplemented with 10% FBS, 1 mM sodium pyruvate (S8636, Sigma-Aldrich), 10 mM HEPES, pH 7.2, 2 mM L-glutamine (25030-024, Gibco, Thermo Fisher Scientific), and 50 µM beta-mercaptoethanol (31350-010, Gibco, Thermo Fisher Scientific). For targeted integration of pre-SH-CDNF, both HEK293 and INS1 parental cells were co-transfected with pcDNA5/FRT/TO pre-SH-CDNF and pOG44 vectors (Invitrogen) using FugeneHD (E2311, Roche, Basel, Switzerland) transfection reagent. Two days after transfection, the stable cell lines were selected with 50 µg/mL Hygromycin-B Gold (ant-hg-1, Invivogen) for 2 weeks. The generation of the HEK293 GFP-SH and INS1 GFP-SH cell lines has been described elsewhere [25].

To check for the doxycycline-induced expression of SH-tagged CDNF, the respective HEK293 pre-SH-CDNF cells were cultivated on 6-well plates and SH-CDNF expression was induced with 1 μg/mL doxycycline (D9891, Sigma-Aldrich) for 24 h before harvesting cells in Laemmli buffer. Equal amounts of cell lysate were used for protein electrophoresis and subsequent immunoblotting with rabbit anti-CDNF antibody (300–100, Icosagen, Tartu, Estonia) and mouse anti-tubulin (T9026, Sigma-Aldrich) and respective IRDye conjugated anti-rabbit and anti-mouse secondary antibodies (LI-COR, Lincoln, NE, USA). Infrared signals were visualized with the Odyssey scanner (LI-COR).

### 4.6. Expression Plasmids for Bimolecular Fluorescence Complementation Assay (BiFC)

The generation of pEZY BiFC Jun-NV, pEZY BiFC Max-CV, pEZY BiFC Fos-CV, pEZY BiFC Grp78-NV, and pEZY BiFC Grp170-NV has been described in detail before [25]. pENTR221 pre-CV-CDNF and pENTR221 pre-NV-CDNF were generated by amplifying the VC155 (C-Venus) and VN173 (N-Venus) sequences from the respective BiFC destination vectors, followed by insertion of these between the sequences coding for the signal peptide (pre-) and mature regions of human CDNF in the pENTR221 CDNF stop. The respective BiFC expression plasmids (pEZY BiFC pre-NV-CDNF and pEZY BiFC pre-CV-CDNF) were made by LR clonase recombination reaction of pENTR221 pre-NV-CDNF and pENTR221 pre-CV-CDNF into the pEZY Myc-His destination vector. The pEZY BiFC pre-CV-CDNF construct was further used as a template to generate pEZY BiFC pre-CV-N-CDNF and pEZY BiFC pre-CV-C-CDNF with 2xPhusion Green HSII HF master mix, following the manufacturer’s recommendations (Thermo Fisher). Briefly, the template was linearized by using inverse PCR. The 5′-phosphorylated primers used for pEZY BiFC pre-CV-N-CDNF were 5′-TGAGACCCAGCTTTCTTGTAC-3′ and 5′-CAGCTCACAGATCTGGCTAT-3′. The 5′-phosphorylated primers used for pEZY BiFC pre-CV-C-CDNF were 5′-AAATATGAAAAAACACTGGACTTGGCATC-3′ and 5′-GCTGCCACCGGATCCACCAG-3′. The template was then digested by DpnI restriction (Thermo Fisher); products were run on an agarose gel, cut, and purified using the NucleoSpin Gel and PCR Clean-Up Mini Kit (Macherey-Nagel, Düren, Germany), followed by ligation using the T4 DNA ligase (Thermo Fisher).

### 4.7. Sympathetic Neuronal Cell Culture and Microinjection

Mouse superior cervical ganglion (SCG) sympathetic neuronal culture and neuron microinjection have been described before [82]. Briefly, SCG neurons of postnatal day 1–2 NMRI strain mice were grown on polyornithine (P4957, Sigma-Aldrich)–laminin (L2020, Sigma-Aldrich)-coated dishes with 30 ng/mL of 2.5 S mouse NGF (G5141, Promega, Madison, WI, USA). On 6 DIV, the nuclei were microinjected with CDNF expression plasmid (10 ng/uL) or with the respective empty plasmid (vector). A reporter plasmid encoding for enhanced green fluorescent protein (EGFP) was co-injected to allow for the identification and counting of successfully injected neurons. For protein microinjection, recombinant full-length (fl) human CDNF protein (P-100-100, Icosagen) in PBS at 200 ng/ul was microinjected directly into the cytoplasm together with fluorescent reporter Dextran Texas Red (MW 70000 Da) (D1830, Invitrogen, Molecular Probes). On average, 50–80 neurons were successfully injected per experimental group. One day after the microinjection, tunicamycin (ab120296, Abcam, Cambridge, United Kingdom) was added at a final concentration of 2 µM. IRE1 signaling inhibitors 4μ8C (4479, Tocris Bioscience, Bristol, UK) or KIRA6 (19151, Cayman Chemical, Ann Arbor, MI, USA) or PERK signaling inhibitor GSK2606414 (516535, Merck Millipore, Burlington, MA, USA) were used when indicated. Three days later, living fluorescent (EGFP-expressing or Dextran Texas Red-containing) neurons were “blindly” counted and expressed as percent of initial living fluorescent neurons counted 2–4 h after microinjection. Survival percentages in CDNF plasmid- or protein-injected groups were compared to the empty vector or PBS-injected controls of the same treatment group using GraphPad Prism (www.graphpad.com, version 8.0.0 or later for Windows or version 9.0.0 or later for Mac OS, San Diego, CA, USA) to perform ordinary one-way analysis of variance (ANOVA) followed by Sidak’s multiple comparison *post hoc* test. N = 3–6 independent experiments were performed. For CDNF immunostaining, SCG neurons were cultured on glass cover slips and microinjected after 7 days in vitro with pCR3.1 plasmid encoding for wt human CDNF. The cells were fixed with 4% PFA 48 h after microinjection, and stained with the following antibodies (1:400 dilution used for all): rabbit anti-CDNF (2.5 μg/mL, 300–100, Icosagen), mouse anti-PDI (2.5 μg/mL, ADI-SPA-891-F, Enzo Life Sciences, Farmingdale, NY, USA), mouse anti-GM130 (0.625 μg/mL, 610823, BD Biosciences, Franklin Lakes, NJ, USA), goat anti-GRP78 (0.5 μg/mL sc-1051, Santa Cruz Biotechnology Inc, Dallas, TX, USA), Alexa Fluor 488 goat anti-rabbit IgG (H+L) (A-11008, Invitrogen), Alexa Fluor 568 donkey anti-goat IgG (H+L) (A-11057, Invitrogen), and Alexa Fluor 568 goat anti-mouse IgG (H+L) (A-11004, Invitrogen). The nuclei were stained with DAPI (D9542, Sigma-Aldrich). The fluorescent image stacks were acquired using the confocal microscope TCS SP5 equipped with LAS AF 1.82 (Leica Microsystems Inc., Wetzlar, Germany). The objective was Leica HCX PL APO x63/1.3 GLYC CORR CS (21 °C). The lasers used were DPSS 561 nm/20 mW, OPSL 488 nm/270 mW, and diode 405 nm/50 mW, with the beam splitter QD 405/488/561/635. The images were deconvoluted by AutoQuant Auto-Deblur 3D Blind Deconvolution software (Media Cybernetics, Rockville, MD, USA). Deconvoluted images were all processed identically with the ImageJ program. Pearson’s coefficient in colocalized volume was calculated by Imaris 9.2.1 software (Bitplane, Belfast, UK). For studying the localization of CDNF in ER-stressed cells, sympathetic neurons were treated with 2 μM tunicamycin for 48 h before being fixed. Staining was repeated n = 3 times, shown as representative images; for Pearson’s colocalization coefficient, 5–7 neurons were used.

### 4.8. Dopamine Neuron Culture, RNA Isolation, Reverse Transcription, and Quantitative PCR

The midbrain floors were dissected from the ventral mesencephalic tissue of 13-day-old NMRI strain mouse embryos. The tissues were incubated with 0.5% trypsin (103139, MP Biomedical) in HBSS (Ca^2+^/Mg^2+^-free) (14170112, Invitrogen) for 20 min at +37 °C, and then mechanically dissociated. Cells were plated onto 96-well plates coated with poly-L-ornithine. Equal volumes of cell suspension (total number of 30 000 cells containing approximately 1200–2000 TH-positive cells) were plated onto the center of the well. The cells were grown in DMEM/F12 medium (21330020, Thermo Fisher Scientific) containing N2 supplement (17502001, Thermo Fisher Scientific) for 5 days, without any neurotrophic factor. Then, the cells were treated with thapsigargin (20 nM) (T7458, Thermo Fisher Scientific) or 6-hydroxydopamine hydrochloride (6-OHDA, 10 μM, H4381, Sigma Aldrich) and CDNF (100 ng/mL). After three days, the neuronal cultures were fixed and stained with anti-tyrosine hydroxylase antibody (MAB318, Millipore Bioscience Research Reagents). Images were acquired by CellInsight high-content imaging equipment. Immunopositive neurons were counted by CellProfiler software, and the data were analyzed by CellProfiler analyst software. Per each condition n = 6–8 independent experiments were performed. Data were analyzed by ordinary one-way analysis of variance (ANOVA) and Sidak’s multiple comparisons post hoc test using GraphPad Prism (version 8.0.0 or later for Windows or version 9.0.0 or later for Mac OS). The results are expressed as % of cell survival compared non-toxin treatment neurons.

For quantitative PCR analysis, midbrain dopaminergic neurons were cultured for 5–7 days and then treated with thapsigargin (200 nM). Recombinant CDNF protein (100 ng/mL) was added to the cultures at the same time. After 24 h, RNA from cultured cells was isolated by TriReagent^®^ (RT118, Molecular Research Center, Cincinnati, OH, USA) according to the manufacturer’s instructions. RNA was reverse-transcribed to cDNA with RevertAid™ Premium Reverse Transcriptase (EP0441, Fermentas UAB, Thermo Fisher Scientific Inc.). Quantitative PCR was performed using the LightCycler^®^ 480 SYBR Green I Master (04887352001, Roche Diagnostics GmbH, Mannheim, Germany) and Roche LightCycler^®^ 480 Real-Time PCR System (Roche Diagnostics GmbH). The expression levels were normalized to the levels of β-actin in the same samples. Primers for *Grp78*, *Chop*, *sXbp1*, *tXbp1*, *Atf4*, *Atf6α* used in quantitative PCR were synthetized using previously published sequences [24,25,79,83]. Each transcript’s levels were normalized to the beta-actin housekeeping gene and presented as a fold change to the respective control data set from non-treated neurons. For each condition, n = 10–18 experiments were performed. GraphPad Prism (version 8.0.0 or later for Windows or version 9.0.0 or later for Mac OS) was used for statistical analysis using repeated-measures ANOVA followed by Sidak’s multiple comparison post hoc test.

### 4.9. Affinity Purification

Affinity purification and subsequent steps of AP-MS were performed as described earlier [25]. Briefly, stable isogenic HEK293 pre-SH-CDNF, HEK293 GFP-SH, INS1 pre-SH-CDNF, and INS1 GFP-SH cells were grown on 15 cm plates. CDNF and GFP expression was induced using 1 µg/mL doxycycline (D9891, Sigma-Aldrich) for 24 h at approximately 70% cell confluency. For each affinity purification sample, cells from five 15 cm plates were washed with ice-cold PBS (containing 0.1 mM MgCl_2_ and 0.1 mM CaCl_2_), harvested in PBS–1 mM EDTA, snap-frozen, and stored at −70 °C until further use. For affinity purification, 3 mL of HENN lysis buffer (50 mM HEPES-NaOH, pH 8.0, 5 mM EDTA, 150 mM NaCl, 50 mM NaF, 0.5% NP40, 1 mM PMSF, 1.5 mM Na_3_VO_4_, and 1× protease inhibitor cocktail (P8340, Sigma-Aldrich)) was used to lyse the frozen pellet on ice for 10 min. Lysates were cleared by two centrifugations at 16,000× *g* for 15 min at 4 °C to remove any insoluble material and loaded onto spin columns (732-6008, Bio-Rad Laboratories, Hercules, CA, USA) containing 200 µL Strep-Tactin beads (2-1201-010, IBA GmbH). Subsequently, the beads were washed three times with 1 mL ice-cold HENN lysis buffer, followed by three washes with 1 mL HENN buffer without detergent and inhibitors. Then, 1 mM of D-biotin (29129, Thermo Fisher Scientific) in HENN buffer without detergent and inhibitors was used to elute protein complexes.

### 4.10. Preparation for Mass Spectrometry Analysis

First, 100 mM ammonium bicarbonate (NH_4_HCO_3_) was used to neutralize the AP eluates. Disulfide bonds were then reduced and alkylated using 5 mM Tris (2-carboxyethyl) phosphine (TCEP) and 10 mM iodoacetamide, respectively. Tryptic peptides were generated by adding 1 μg trypsin (Promega, Fitchburg, WI, USA) per pull-down sample. After overnight incubation at 37 °C, samples were quenched with 10% trifluoroacetic acid (TFA) and purified using C18 Micro Spin Columns, following the manufacturer’s instructions (The Nest Group, Southborough, MA, USA). Finally, for LS-MS analysis, samples were re-dissolved in 30 μl buffer A (0.1% trifluoroacetic acid and 1% acetonitrile in LC-MS-grade water).

### 4.11. Mass Spectrometry Analysis

An Orbitrap Elite hybrid mass spectrometer coupled to an EASY-nLC II-system using the Xcalibur version 2.7.0 SP1 (Thermo Fisher Scientific) was used for LC-MS analysis. First, 4 μL of the tryptic peptide mixture was loaded onto a C18-packed precolumn (EASY-Column™ 2 cm × 100 μm, 5 μm, 120 Å, Thermo Fisher Scientific) in 10 μL of buffer A. Then, the sample was loaded onto the C18-packed analytical column (EASY-Column™ 10 cm × 75 μm, 3 μm, 120 Å, Thermo Fisher Scientific). A 60-min linear gradient at a constant flow rate of 300 nL/min from 5 to 35% of buffer B (98% acetonitrile and 0.1% formic acid in MS-grade water) was used to separate the peptides. Data-dependent acquisition was used for analysis: one high-resolution (60,000) FTMS full scan (*m*/*z* 300–1700) followed by top20 CID-MS2 scans in ion trap (energy 35). Maximum FTMS fill time was 200 ms (Full AGC target 1,000,000) and the maximum fill time for the ion trap was 200 ms (MSn AGC target of 50,000). Precursor ions with more than 500 ion counts were allowed for MS. Scan preview mode was used to enable the high resolution in FTMS.

### 4.12. Protein Identification and Quantification

A combination of the Andromeda search engine and MaxQuant proteomics software [84] was used for protein identification and MS1 quantification. Human or rat components of the UniProt database complemented with trypsin, BSA, GFP, and tag sequences (human: release 2016_1; 20,149 entries, rat: release 2016_10; 7973 entries) were used to search Thermo raw files. The rat component of the database was additionally completed with the human CDNF sequence. Carbamidomethylation (+57.021464 Da) of cysteine residues was used as static modification and oxidation (+15.994491 Da) of methionine as dynamic modification. Error tolerances on the precursor and fragment ions were ±4.5 ppm and ±0.5 Da, respectively. Peptide false discovery rate (FDR) was set to <0.05.

### 4.13. MS Data Filtering and Analysis

MS data filtering and subsequent analysis was done essentially as described before [25]. The Perseus software platform (version 1.5.5.3, Marcus Hook, PA, USA) [85] was used for data filtering and analysis. Intensity values from MaxQuant software were base2 logarithmized for normal distribution. Proteins identified in three or less runs of the quadruplicate runs per bait were not included in further analysis. Random selection from a normal distribution shifted from the measured data distribution towards the lower intensity values (down shift 1.8 and width 0.3 of standard deviations) was used for imputing missing values. Ribosomal and keratin proteins were manually excluded from further analysis. For visualizing and filtering high-confidence interactors in Perseus, we used the volcano plot. The *p* values were adjusted to 0.1% FDR and the scaling factor s0 was set to 2. We used the PINA database (https://omics.bjcancer.org/pina/, accessed on 19 June 2018) [86,87] to identify interactions within our data sets. The combined networks of PINA output and the interactions identified in this study were visualized using Cytoscape software (version 3.5.1, Boston, MA, USA) [88]. For overrepresentation analysis, we used the Panther database (version 11, accessed 17 November 2020) [89].

### 4.14. Bimolecular Fluorescence Complementation Assay

HEK293 cells were plated onto poly-D-lysine (P0899, Sigma-Aldrich)-coated glass cover slips 48 h before transfection. JetPEI transfection reagent (101, Polyplus Transfection, Illkirch, France) was used according to the manufacturer’s instructions to co-transfect cells with the indicated pEZY BiFC C-Venus and N-Venus plasmids. The cells were fixed with 4% PFA, permeabilized with 0.1% Triton X-100, and stained with the indicated organellar marker antibodies approximately 20–24 h after transfection. Nuclear counterstaining was done using Hoechst33342 (H1399, Invitrogen) and ProLong™ Diamond Antifade Mountant (P36965, Thermo Fisher Scientific) was used for mounting.

### 4.15. Imaging

All microscopy images were taken using the LSM 700 (Carl Zeiss) confocal microscope, LCI Plan-Neofluar 63x/1.30 glycerol immersion objective, at room temperature, and Zen Black 2.1 acquisition software (version 11.0.0.190, Carl Zeiss AG, Oberkochen, Germany). Image analysis was done using the Zen Blue Lite (version 2.0, Carl Zeiss AG, Oberkochen, Germany), PHOTO-PAINT, and CorelDraw programs from the CorelDRAW Graphics Suite 2017. Post-imaging processing was done using Corel PHOTO-PAINT 2017 (Corel Corporation, Ottawa, ON, Canada) using the brightness/contrast/intensity adjustment settings equally and simultaneously for images from the same imaging series.

## 5. Conclusions

CDNF is a neurotrophic factor that has recently been tested in combined phase I/II clinical trials for the treatment of Parkinson’s disease—a progressive neurodegenerative disorder characterized by the loss of midbrain dopamine neurons. However, its biology and mechanism of action are poorly understood. In the current paper, we shed light on the molecular mechanisms of action of this protein belonging to a MANF/CDNF family of neurotrophic factors. We show that CDNF localizes to the endoplasmic reticulum and is upregulated in several tissues after the systemic administration of ER stress-inducing tunicamycin in vivo. Moreover, CDNF rescues cultured sympathetic and dopamine neurons from ER stress-related apoptosis through regulating unfolded protein response signaling pathways. We report that CDNF interacts with several proteins involved in the maintenance of ER homeostasis. To conclude, our findings place CDNF amongst the ER stress and unfolded protein response regulators and allow us to suggest that CDNF exerts its antiapoptotic function through interactions with UPR pathways, steering cells towards survival.

## Figures and Tables

**Figure 1 ijms-23-09489-f001:**
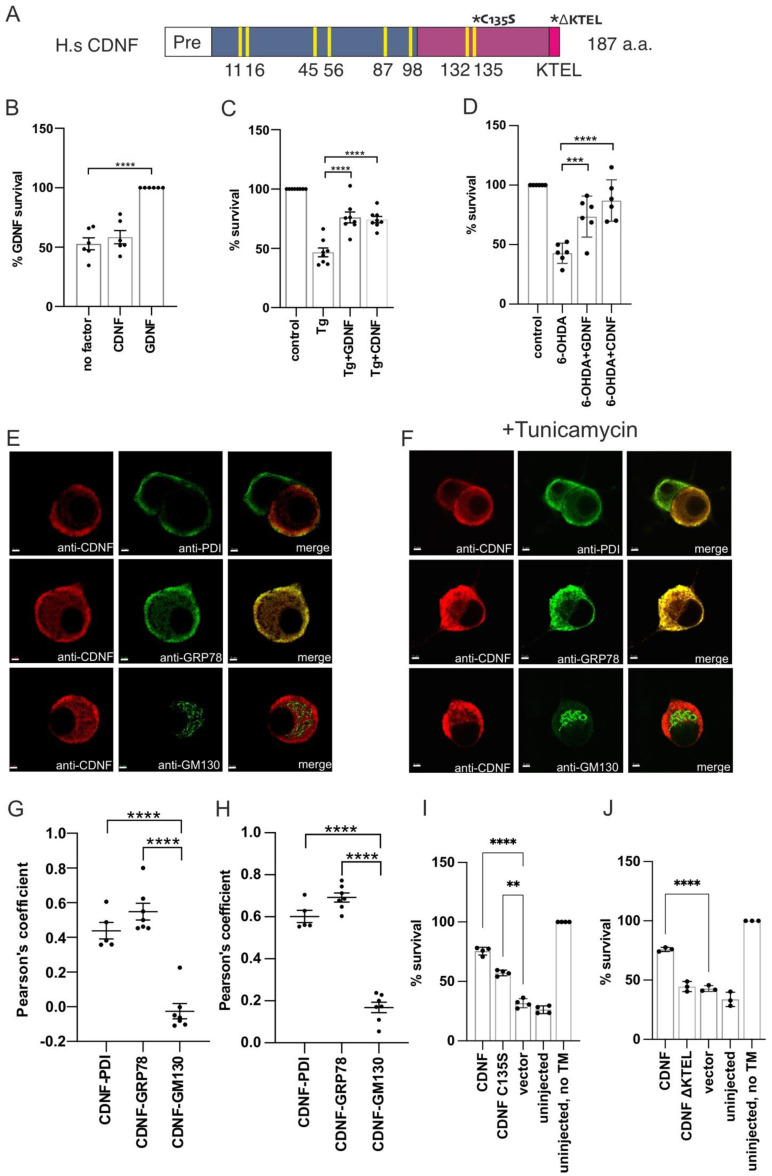
CDNF is an ER–localized protein that promotes survival of ER-stressed neurons in vitro. (**A**) Schematic primary structure of human CDNF. The ER-targeting (Pre) sequence is marked as white, the N-terminal domain as blue, the C-terminal domain as mauve, and the very C-terminal KDEL-like ER retention signal (KTEL) as bright pink. Yellow bars mark the conserved cysteine residues, numbered according to the mature protein. Asterisks mark CDNF mutations used in this study. Pro-CDNF is 187 amino acids long. CDNF numbering starts from the N-terminus on mature CDNF. (**B**) Midbrain dopamine (DA) neuron cultures from E13.5 mice were cultured in the presence of CDNF or GDNF at 100 ng/mL or culture media without neurotrophic compound (no factor) for 5 days. DA neurons were identified by tyrosine hydroxylase (TH) immunostaining and expressed as % of cell survival in each condition compared to GDNF-treated neurons. Shown are the means ± S.D. of 6 independent experiments per condition. Data were analyzed by ordinary one-way analysis of variance (ANOVA) and Sidak’s multiple comparisons *post hoc* test. (**C**) E13.5 mouse midbrain floor cultures were treated with 100 nM thapsigargin (Tg) only or Tg and the indicated growth factors from DIV 5 to DIV 8. After three days TH-positive neurons were counted and expressed as a percentage of non-Tg-treated neurons. Shown are the means of 8 experiments ± S.D. Tg-treated group was compared to either Tg+CDNF or Tg+GDNF group using ordinary ANOVA and Sidak’s multiple comparison *post hoc* test. (**D**) E13.5 mouse midbrain DA cultures were treated with 10 μM 6-OHDA only or 6-OHDA and the indicated growth factors from DIV 5 to DIV 8. After three days, TH-positive neurons were counted and expressed as a percentage of non-6-OHDA treated neurons. Shown are the means of 6 experiments ± S.D. The 6-OHDA-treated group was compared to either 6-OHDA+CDNF or 6-OHDA+GDNF group using ordinary ANOVA and Sidak’s multiple comparison *post hoc* test. (**E**,**F**) CDNF localization in mouse superior cervical ganglion (SCG) neurons by immunofluorescence. Neurons were microinjected with *CDNF* expression plasmid (**E**) and treated with 2 µM tunicamycin (**F**). CDNF protein colocalized with ER markers PDI and GRP78 but not with the Golgi marker GM130. Pearson’s colocalization coefficients were calculated from randomly selected 5–7 individual neurons per indicated immunostaining pair from E (**G**) and F (**H**) using Imaris x64 software and analyzed by ordinary one-way ANOVA. Scale bar is 3 μm. E, embryonic day; DIV, days in vitro. (**I**,**J**) SCG neurons isolated on postnatal day 1 or 2 were injected with plasmid vector encoding CDNF or indicated CDNF mutants and treated with 2 µM tunicamycin to induce ER stress-related apoptosis. The number of living injected, GFP-positive neurons was calculated 72 h after the injections and expressed as the percentage of initially injected neurons. Shown are the means of 3 experiments ± S.D. *CDNF* plasmid-injected groups were compared to the empty vector using one-way ANOVA and Sidak’s multiple comparison *post hoc* test. *, **, ***, and **** denote *p* < 0.05, *p* < 0.01, *p* < 0.001, and *p* < 0.0001. The null hypothesis was rejected at *p* < 0.05.

**Figure 2 ijms-23-09489-f002:**
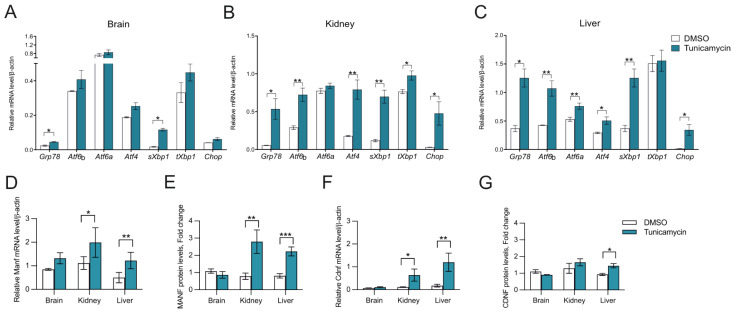
CDNF expression is upregulated after experimentally induced ER stress in mouse tissues in vivo. Increased expression of UPR genes in (**A**) brain, (**B**) kidney, and (**C**) liver tissues of mice injected with tunicamycin dissolved in dimethylsulfoxide (DMSO) and diluted in PBS in contrast to control (DMSO) + PBS-injected mice analyzed by RT-qPCR. n = 2–5 per group. (**D**) MANF mRNA and (**E**) protein levels and (**F**) CDNF mRNA and (**G**) protein levels in brain, kidney, and liver analyzed by RT-qPCR and ELISA from DMSO-injected control mice and tunicamycin-injected mice. n = 2–5 mice per group. Data are presented as mean ± SEM. Student’s t-test, * *p* < 0.05, ** *p* < 0.01, *** *p* < 0.001.

**Figure 3 ijms-23-09489-f003:**
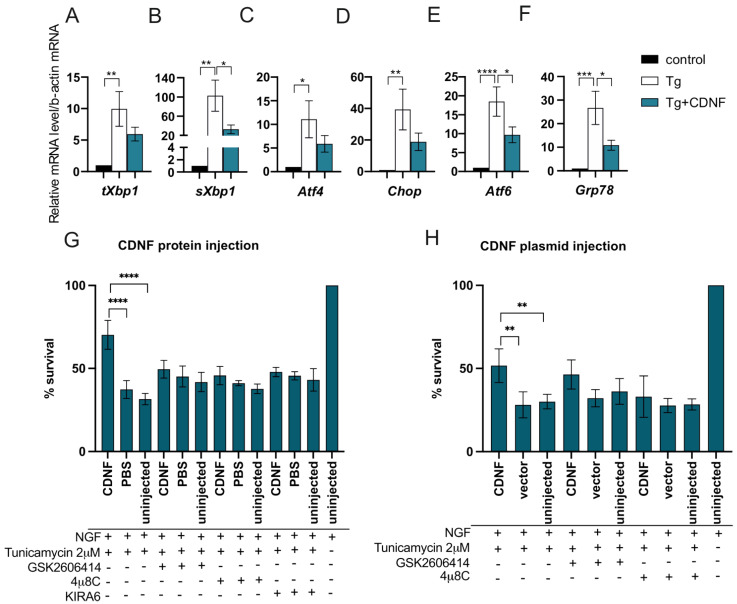
CDNF promotes survival of ER–stressed cultured neurons and regulates UPR signaling in vitro. (**A**–**F**) CDNF treatment downregulated the expression of UPR genes *sXbp1*, *Atf6*, and *Grp78* in embryonic mouse DA neurons under thapsigargin (Tg)-induced ER stress. DA neurons were cultured 5–7 days before adding CDNF (100 ng/mL) and inducing ER stress by adding 200 nM Tg. RNA was isolated from these neurons after 24 h. The expression levels of UPR marker transcripts were determined by qPCR. Each transcript’s levels have been normalized to beta-actin housekeeping gene and presented as a fold change to respective control data set from non-treated neurons. Shown are means of n = 10–18 experiments ± S.E.M. Repeated-measures ANOVA and Sidak’s multiple comparison post hoc test. (**G**,**H**) Survival-promoting activity of CDNF against ER stress is dependent on active IRE1α and PERK pathways. Mouse SCG neurons were microinjected with (**G**) recombinant human CDNF protein or (**H**) CDNF expression plasmid and treated with 2 µM tunicamycin and 2 µM PERK signaling inhibitor GSK2606414 or 25 µM IRE1α signaling inhibitor 4μ8C or 2 µM IRE1α inhibitor KIRA6. Then, 72 hours later, the number of living injected, fluorescent neurons was counted and expressed as the percentage of initially injected neurons. Shown are the means of 3–6 experiments ± S.D. Survival percentages in CDNF plasmid- or protein-injected groups were compared to the empty vector or PBS-injected controls of the same treatment group using ordinary one-way analysis of variance (ANOVA) and Sidak’s multiple comparison *post hoc* test. *, **, ***, **** denote *p* < 0.05, *p* < 0.01, *p* < 0.001, *p* < 0.0001, respectively.

**Figure 4 ijms-23-09489-f004:**
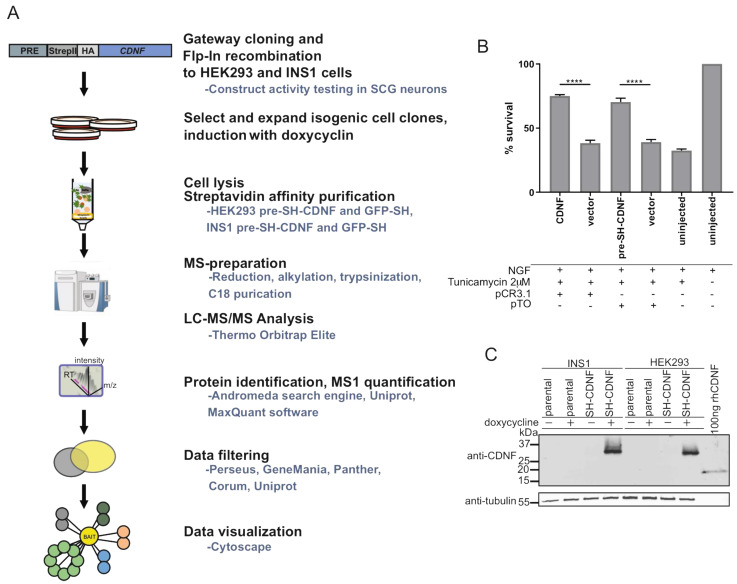
Generation of doxycycline–inducible HEK293 and INS1 cell lines for AP–MS. (**A**) Schematic pipeline of the AP-MS. Signal peptide (PRE), StrepII, and HA tags are indicated. (**B**) CDNF expressed from pre-SH-CDNF construct is biologically active. Nerve growth factor (NGF)-maintained mouse embryonic SCG neurons were injected with the indicated expression plasmids and treated with 2 µM tunicamycin. The number of surviving injected neurons was expressed as % of the number of living injected neurons counted 3–4 h after injection. Shown are the means of five independent experiments ± S.E.M. Data of each experimental group were compared to respective empty vector control using ordinary ANOVA and Sidak’s multiple comparison post hoc test. **** denotes *p* < 0.0001. The null hypothesis was rejected at *p* < 0.05. (**C**) Expression of SH-tagged CDNF upon doxycycline induction in Hek293 and INS1 cell lines. Whole-cell lysates of HEK293 and INS1 parental and CDNF-expressing (SH-CDNF) cell lines were analyzed by immunoblotting using the indicated antibodies. Recombinant human CDNF was used as an antibody control.

**Figure 5 ijms-23-09489-f005:**
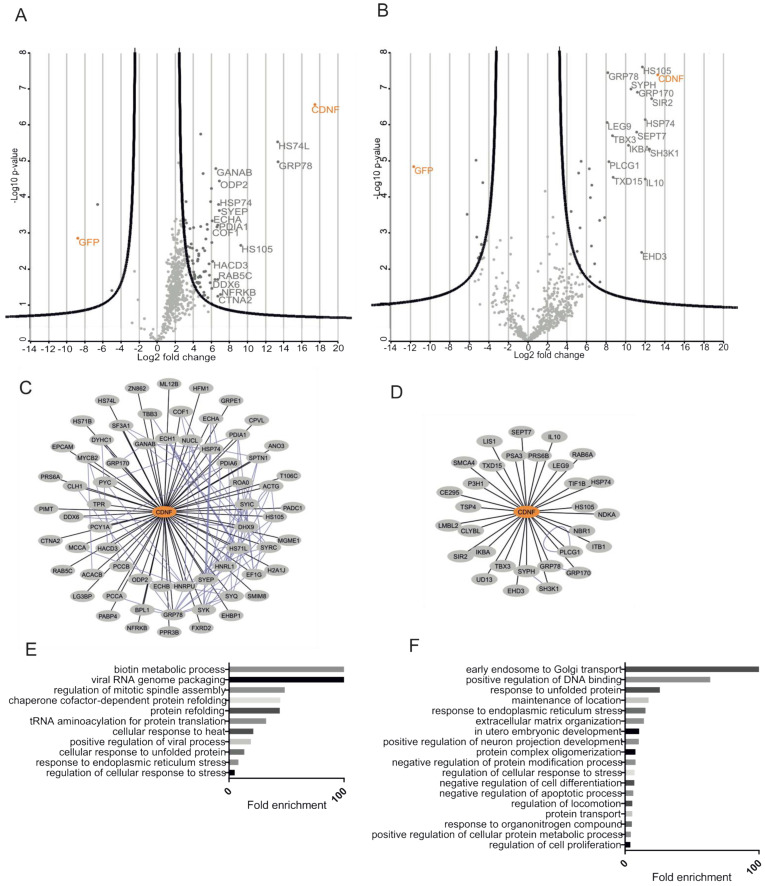
CDNF interactome in HEK293 and INS1 cell lines expressing SH–CDNF. Volcano plots of protein abundance (log2 fold change) against the t-statistic significance in (**A**) HEK293 cell line and (**B**) INS1 cell line. AP-MS from the respective GFP-SH cell lines served as negative controls. CDNF interactomes consisting of significantly enriched proteins are shown on the right side of the respective parabola and were calculated using the volcano plot plugin of Perseus software package (FDR ≤ 0.001, S0 = 2). Bait names (CDNF and GFP) are shown in orange. The UniProt entry names of the 15 most enriched proteins of each interactome are shown. Interaction network map of CDNF in (**C**) HEK293 cell line and (**D**) INS1 cell line. The network was supplemented with node–node interaction data from PINA2 database (blue lines). Gene Ontology (GO) Biological Process (BP) overrepresentation analysis of CDNF interactome from (**E**) HEK293 cell line and (**F**) INS1 cell line. The analysis was done using the Panther online software. Significantly overrepresented (Bonferroni-corrected *p*-value < 0.05) terms are listed in the order of decreasing fold enrichment values. Terms with >100 fold enrichment are shown as 100-fold enrichment.

**Figure 6 ijms-23-09489-f006:**
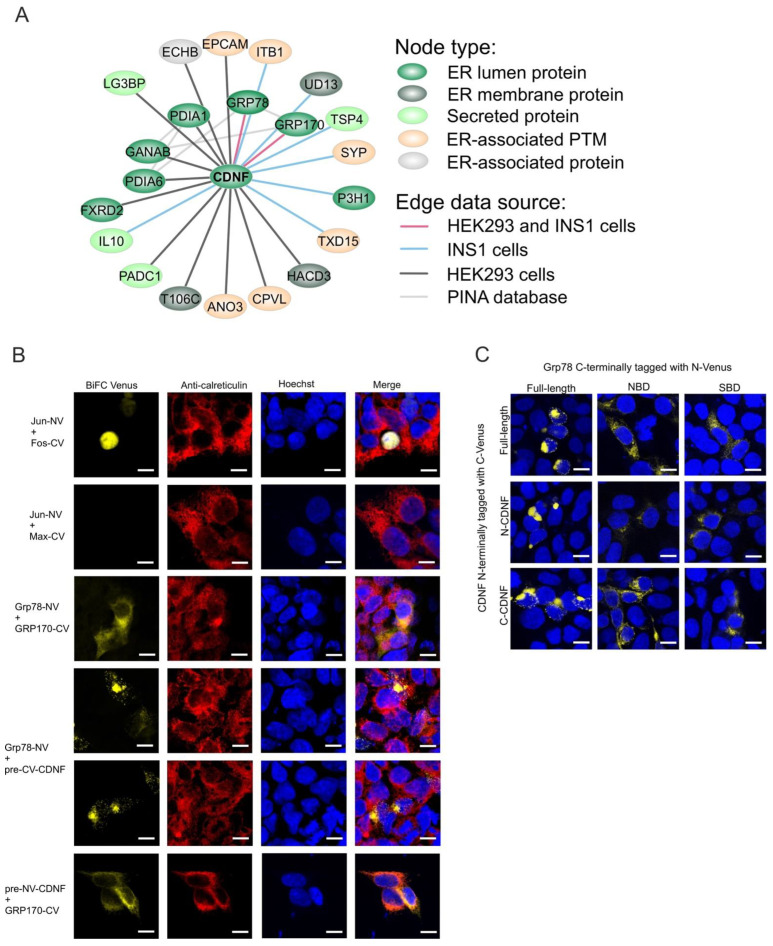
The ER-localized interactome of CDNF. (**A**) An interaction map consisting of a subset of ER-localized or ER-associated proteins in the CDNF interactome from both HEK293 and INS1 cell lines. Node color coding represents the subcellular location in respect to ER of each node based on its respective UniProtKB database entry. Edge color represents the origin of the respective interaction data. (**B**) CDNF interaction with GRP78 and GRP170 in HEK293 cells by bimolecular fluorescence complementation (BiFC) assay. Yellow fluorescent signal indicates interaction between the tested proteins. Jun-NV+Fos-CV and GRP78-NV+GRP170-CV pairs were used as positive controls as BiFC signal formation in the nucleus and ER, respectively. Jun-NV+Max-CV pair was used as a negative control for signal formation. ER was visualized by anti-calreticulin immunostaining (red) and nuclei by Hoechst staining (blue). (**C**) HEK293 cells were co-transfected with the indicated CDNF and GRP78 BiFC plasmids and nuclei were visualized by Hoechst staining. White scale bars denote 10 µM. NBD, nucleotide-binding domain; SBD, substrate-binding domain of GRP78. Shown are representative images from n = 3–5 independent experiments per binding pair.

## Data Availability

Data are contained within the article.

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
