# Peer review of "CDNF Interacts with ER Chaperones and Requires UPR Sensors to Promote Neuronal Survival"

_ijms, 2022, doi:10.3390/ijms23169489_

Round 1

Reviewer 1 Report

The manuscript titled “CDNF interacts with ER chaperones and requires UPR sensors to promote neuronal survival” by Eesmaa, A.; et al. is an original scientific work where the authors combine numerous complementary techniques like RT-qPCR, ELISA, immunobloting, bimolecular fluorescence complementation assays and affinity purification coupled to mass spectrometry which makes this study really robust and serves to assess the underpinning molecular mechanisms of cerebral dopamine neurotrophic factor (CDNF) and its regulative response upon stress factors. The conducted research is highly innovative combining the above described approaches and could shed light to those action mechanisms related to neuronal diseases like Parkinson’s disease and ultimately, conduct clinical trials biased to the molecular targets encountered in this scientific work. The gathered findings may be relevant for the examined field. The results achieved are well-discussed during the main body of the reported manuscript. The scientific paper is well written. In my opinion the present manuscript is innovative and the methodological approached used matches with the scope of International Journal of Molecular Sciences. For the above described reasons, I recommend the publication in International Journal of Molecular Sciences once the following remarks will be fixed:

--------

ABSTRACT

Authors should introduce the term “mesencephalic astrocyte-derived neurotrophic factor” before its abbreviation “(…) member of the CDNF/MANF family” (line 11).

--------

KEYWORDS

The selected keywords are the appropriated respecting to the submitted work. Nevertheless, I may exchange “protein-protein interaction study” by “protein-protein interactions”. Eventually, in literature this term is abbreviated as PPI as authors indicate in line 312.

--------

INTRODUCTION

Introduction section is clear and concise. Furthermore, this section provides accurate information with relevant references of the system of study. Authors should take care of the way the bibliography citations are inserted. Some examples are the following (but it exists many others): “(…) [1]-[6].” (line 32) or “(…) [10], [11]” should be modified by “(…) [1-6]” and “(…) [10,11]”, respectively.

Author’s group is a well-recognized team in this field, for this reason I may see appropriate introduce the following review work in the Introduction section:

[1] Lindholm, P.; et al. Cerebral dopamine neurotrophic factor protects and repairs dopamine neurons by novel mechanism. Mol. Psychiatry 2022, 27, 1310-1321. https://doi.org/10.1038/s41380-021-01394-6.

Finally, it may be also convenient to introduce a small statement indicating that it has been deeply reported that neurotrophic factors can cross the blood-brain barrier and delivered directly to certain brain regions [2].

[2] Kadry, H.; et al. A blood-brain barrier overview on structure, function, impairment, and biomarkers of integrity. Fluids Barriers CNS 2020, 17, 69. https://doi.org/10.1186/s12987-020-00230-3.

--------

RESULTS

The most significant outcomes are perfectly explained for all potential target audiences and other stakeholders. Nevertheless, the following points should be addressed:

I)          Figure 1 (line 103). Please, add the scale bars on the images from the fluorescence assays (Fig. 1E and Fig. 1F, respectively).

II)       Authors used dimethyl sulfoxide (DMSO) as negative control to test the effect of tunicamycin in the inyected mouses (Figure 2, line 232). It has been reported that DMSO can induce oxidative stress and cytotoxicity [3]. Authors should add a brief statement on this regard.

[3] Dludla, P.V.; et al. Chapter 25 – The impact of dimethyl sulfoxide on oxidative stress and cytotoxicity in various experimental models. Toxicology Oxidative Stress and Dietary Antioxidants 2021, 243-261. https://doi.org/10.1016/B978-0-12-819092-0.00025-X.

III)    Figure 6 (line 430). Authors should indicate the scale bars dimension for fluorescence assays images analogously to Figure 1.

--------

MATERIALS AND METHODS

Authors have fully explained all the required information to carry out the same type of experiments in other labs from any world-wide research institution. Analysis of variance (ANOVA) is explained during Figure 1, Figure 3 and Figure 4 captions, respectively. Authors should add slightly more details about the statistical analysis carried out and the population size (N) taken into account in the respective Materials & Methods section.

--------

DISCUSSION

In general terms, the article are well-discussed and the cited reference are appropriate. Authors should present some alternative techniques to assess protein-protein interactions like protein fragmentation complementation [4], protein microarrays [5], bioluminescence resonance energy transfer (BRET) [6], förster resonance energy transfer (FRET) [7], fluorescence cross-correlation spectroscopy (FCCS) [8], or single molecule technologies, such as atomic force microscopy based on force spectroscopy (AFM-FS) [9], molecular recognition imaging [10] and optical tweezers (OT) [11].

[4] Blaszczak, E.; et al. Protein-fragment complementation assays for large-scale analysis of protein-protein interactions. Biochem. Soc. Trans. 2021, 49, 1337-1348. https://doi.org/10.1042/BST20201058.

[5] Zhang, L.; et al. Exploration of the HAS-miR-1587-Protein Interaction and the Inhibition to CASK. Int. J. Mol. Sci. 2021, 22, 10716. https://doi.org/10.3390/ijms221910716.

[6] Verweij, E.W.E.; et al. BRET-Based Biosensors to Measure Agonist Efficacies in Histamine H1 Receptor-Mediated G Protein Activation, Signaling and Interactions with GRKs and β-Arrestins. Int. J. Mol. Sci. 2022, 23, 3184. https://doi.org/10.3390/ijms23063184.

[7] Liao, J.; et al. Quantitative FRET (qFRET) Technology for the Determination of Protein-Protein Interaction Affinity in Solution. Molecules 2021, 26, 6339. https://doi.org/10.3390/molecules26216339.

[8] Jakobowska, I.; et al. Fluorescence Cross-Correlation Spectroscopy Yields True Affinity and Binding Kinetics of Plasmodium Lactate Transport Inhibitors. Pharmaceuticals 2021, 14, 757. https://doi.org/10.3390/ph14080757.

[9] Pérez-Domínguez, S.; et al. Nanomechanical Study of Enzyme: Coenzyme Complexes: Bipartite Sites in Plastidic Ferredoxin-NADP+ Reductase for the Interaction with NADP. Antioxidants 2022, 11, 537. https://doi.org/10.3390/antiox11030537.

[10] Marcuello, C.; et al. Molecular Recognition of Proteins through Quantitative Force Maps at Single Molecule Level. Biomolecules 2022, 12, 594. https://doi.org/10.3390/biom12040594.

[11] Choudhary, D.; et al. Bio-Molecular Applications of Recent Developments in Optical Tweezers. Biomolecules 2019, 9, 23. https://doi.org/10.3390/biom9010023.

--------

CONCLUSIONS

Even if Conclusions section is optional, I may encourage the authors to prepare a small statement with the most relevant outcomes found in the present scientific work. This point may increase the impact of the present submitted manuscript.

--------

REFERENCES

Bibliography citations are not in the proper format of International Journal of Molecular Sciences. Authors should take care of this aspect.

--------

OVERVIEW AND FINAL COMMENTS

The submitted work is well-designed and the gathered results are interesting for the design and fabrication of next-generation of therapies against neurodegenerative disorders. For this reason, I will recommend the present scientific manuscript for further publication in International Journal of Molecular Sciences once all the aforementioned suggestions will be properly fixed.

Reviewer 2 Report

In this manuscript, the authors propose a new perspective on CDNF that is specifically helpful for further understanding of Parkinson disease. This does contribute academically. However, I still make the following review comments based on academic requirements.

1.      In line 27-29, the author makes the following description:

was first characterized as a trophic factor for midbrain dopamine neurons in a rat 6-hydroxydopamine (6-OHDA) model of Parkinson’s disease [1].

I suggest that the authors add the date of the literature to this description. Because when there are important discoveries, the presentation of the age is important.

2.      In line 67-68, the author makes the following description:

It has been suggested that CDNF too, is participating in the regulation of UPR signaling.

Please provide supporting evidence from the literature.

3.      In line 149-153, the author makes the following description:

Furthermore, we show that inhibiting IRE1α and PERK pathways of UPR blocks CDNF anti-apoptotic activity. Additionally, we performed the first protein-protein interaction screening for CDNF, and report novel protein partners of CDNF. We suggest that CDNF exerts its cytoprotective properties by modulating UPR signaling towards cell-survival responses.

This description is mainly to illustrate the research purpose of this manuscript. But the authors describe it like "Conclusion" rather than "Research purpose". I suggest that the authors make appropriate corrections.

4.      In Figure 1G-H, the authors adopt the analysis strategy of Pearson’s coefficient. The authors are asked to justify this analytical strategy, as there is a limit to what statistical significance the correlation coefficient can present.

5.      In Figure 2, the performance of brain is different from kidney and liver. This is an interesting result. Unfortunately, the authors did not discuss this result. I suggest that authors can make additions.

6.      In Figure 5A-B, the authors are asked to explain the reasons for treating the values on the horizontal and vertical axes as log2 and -log10.

7.      I suggest that the authors add a description of future research proposals in the “Discussion” section.

8.      Significant "p" worth expressions should be lowercase and italicized. Authors are invited to make uniform corrections regarding the expression of p-values throughout the manuscript.

Reviewer 3 Report

The concerns associated with the manuscript have been listed here:

1) Some of the references are missing from the introduction. The authors have used statements to sound like scientific facts without referencing appropriate literature. Example: While giving the example of pancreatic beta cells as site for high ER stress as a consequence of high amounts of insulin being produced. The authors are advised to carefully add references throughout the introduction where appropriate. 

2) The figure legends should include information for if the experiments have been repeated (experimental repeats). Currently for some of the immunostaining experiments it seems like the experiment just comprises a sample size of 5-7 without the entire staining being repeated for consistency and validation?

3) The entire premise of the paper is built on the importance of CDNF in promoting neuronal survival. However, for the affinity purification and mass spectrometry experiments, the authors chose HEK293 ad INS1 cells. The authors need to better discuss their selection, the potential caveats associated with the selection and the possibility for non-specific pathways being identified as a consequence of the current approach. 

4) The scale bars in Figure 1E and F are barely visible. The panels in Figure 6B for pre-NV-CDNF and GRP170-CV have been switch between Hoechst and Merge. Another question related to Figure 6B is why is the BiFC fluorescence seen in all cells in the field of view when applicable. Please provide an explanation for the same and also provide the calibration bars for the intensities in each channel. 

5) Please discuss the issues and caveats plausible with the approach adopted in section 2.8. Especially since all the constructs contain the ER signal sequence and the C-terminal ER retention sequence. It might not be the most appropriate approach to identify the preferential interaction site domains. Have the authors though of using a more specific approach such as Proximity Ligation Assay to better narrow down the regions based on antibody binding approaches?

6) The concentrations for the antibodies used should be reported in the methods section. 

Reviewer 4 Report

The article by Eesmaa et al. entitled “CDNF interacts with ER chaperones and requires UPRsensors3 to promote neuronal survival” is quite interesting and however, suggests fixing the following issue.

1.     The author must write the aim and methods of the study in the abstract. As per the journal’s instructions, the authors describe briefly the main methods or treatments applied. Include any relevant preregistration numbers, and species and strains of any animals used.

2.     There were no clear blots or images found in the supplementary files.

Round 2

Reviewer 2 Report

The authors' manuscript has been revised as necessary to meet the academic requirements. And the authors contribute to the research of CDNF.